

# Driving factors of aerosol acidity: a new hierarchical quantitative analysis framework and its application in Changzhou, China

Xiaolin Duan[1,#], Guangjie Zheng[1,#,*], Chuchu Chen[1,*], Qiang Zhang[2,*], Kebin He[1]

5    [1] State Key Joint Laboratory of Environmental Simulation and Pollution Control, School of Environment, Tsinghua University, Beijing 100084, China

[2] Ministry of Education Key Laboratory for Earth System Modeling, Department of Earth System Science, Tsinghua University, Beijing, China.

[#] These authors contributed equally.

10   *Correspondence to*: G. Zheng (zgj123@mail.tsinghua.edu.cn); C. Chen (chencc3@mail.tsinghua.edu.cn); Q. Zhang (qiangzhang@tsinghua.edu.cn)

**Abstract.** Aerosol acidity (or pH) plays a crucial role in atmospheric chemistry, influencing the interaction of air pollutants with ecosystems and climate. Aerosol pH shows large temporal variations, while the driving factors of chemical profiles versus meteorological conditions are not fully understood due to the intrinsic complexity. Here, 15 we propose a new framework to quantify the factor importance, which incorporated interpretive structural modelling approach (ISM) and time series analysis. Especially, a hierarchical influencing factor relationship is established based on the multiphase buffer theory with ISM. Long-term (2018 to 2023) observation dataset in Changzhou, China is analyzed with this framework. We found the pH temporal variation is dominated by the seasonal and random variations, while the long-term pH trend varies little despite the large emission changes. This 20 is an overall effect of decreasing $PM_{2.5}$, increasing temperature, and increased alkali-to-acid ratios. Temperature is the controlling factor of pH seasonal variations, through influencing the multiphase effective acid dissociation constant $K_a^*$, non-ideality $c_{ni}$ and gas-particle partitioning. Random variations are dominated by the aerosol water contents through $K_a^*$ and chemical profiles through $c_{ni}$. This framework provides quantitative understanding on the driving factors of aerosol acidity at different levels, which is important in acidity-related process studies and 25 policy-making.

**Short summary:** Aerosol acidity is an important parameter in atmospheric chemistry, while its driving factors, especially chemical profiles versus meteorological conditions, are not yet fully understood. Here, we established a hierarchical quantitative analysis framework to understand the driving factors of aerosol acidity on different time scales. Its application in Changzhou, China revealed distinct driving factors and corresponding mechanisms 30 of aerosol acidity from annual trends to random residues.



## 1. Introduction

Aerosol acidity strongly influences particle mass and chemical constituents by regulating thermodynamic and chemical kinetic processes (Cheng et al., 2016; Pye et al., 2020; Su et al., 2020; Tilgner et al., 2021; Zheng et al., 2020). It is therefore an important parameter in the atmosphere for assessing the impact of atmospheric aerosols

on human health, ecosystems and climate (Nenes et al., 2021; Pye et al., 2020). As direct measurements of aerosol pH in real atmosphere remain unavailable, thermodynamic models are widely adopted to estimate aerosol acidity and investigate its influencing factors (Clegg et al., 2001; Fountoukis and Nenes, 2007; Tao and Murphy, 2021; Zaveri et al., 2008; Zuend et al., 2008).

Driving factors of aerosol pH, especially the relative importance of chemical profiles versus meteorological

conditions, have been widely investigated but still not fully understood. For example, based on long-term observations at six Canadian sites, Tao et al. (Tao and Murphy, 2019) found that temperature largely regulates the aerosol pH in summer, which the chemical profiles may also play a role in winter. Ding et al.(Ding et al., 2019) employed controlled variable tests with the thermodynamic model and concluded that in the North China Plain, sulfate, total ammonia and temperature are the common drivers of pH variations, while total nitrate barely

influence the pH. In comparison, Zhou et al. (Zhou et al., 2022) demonstrated that in the Yangtze River Delta region, non-volatile cations (NVCs, including $Na^+$, $Ca^{2+}$, $K^+$ and $Mg^{2+}$) and sulfate are crucial for annual pH trends, while the seasonal and diurnal variations are determined by meteorological conditions of temperature and RH. Nevertheless, an in-depth investigation into the underlying mechanisms and quantitative attributions of how the meteorology or chemical compositions would influence the aerosol pH is still lacking.

Following the Air Pollution Prevention and Control Action Plan in 2013, the Chinese government continued to introduce the Three-Year Action Plan to Fight Air Pollution (hereinafter referred to as the Action Plan) in 2018. With the implementation of these action plans, both the $PM_{2.5}$ concentration and its chemical components have changed considerably (Bae et al., 2023; Nah et al., 2023; Zhang et al., 2022), which in turn affects aerosol pH. Variation in pH would influence the formations of $PM_{2.5}$ via affecting the gas-particle partitioning of semi-volatile

species (e.g. $HNO_3$) and chemical kinetics, thereby feeding back into the air quality, climate and human health (Cheng et al., 2016; Li et al., 2017; Pye et al., 2020; Su et al., 2020).

The recently proposed multiphase buffer theory offers a new quantitative insight into the aforementioned issue, which shows how and why the chemical profiles and meteorological parameters would influence the aerosol acidity(Zheng et al., 2020, 2022a, 2024b). Here, we established a hierarchical influencing factor relationship of





aerosol pH based on the multiphase buffer theory with interpretive structural modelling approach (ISM). Combining this model with time series analysis, we proposed a new hierarchical quantitative analysis framework to quantify driving factors, and applied it to the long-term observations in Changzhou, China. Distinct driving factors were found for pH variations across different time series components, the underlying mechanisms were quantified, and future implications were also discussed.

## 2. Methods

### 2.1 Ambient measurements and aerosol acidity prediction

Long-term observations of aerosol chemical components and precursor gases are conducted at an urban site of Changzhou Environmental Monitoring Center (31.76° N, 119.96° E), which is located in Changzhou, an important city in the center of Yangtze River Delta (YRD) region. Further details regarding the sampling site and instruments 70 information are described elsewhere(Li et al., 2023; Yi et al., 2022). Briefly, the $PM_{2.5}$ is measured by a Continuous Particulate Matter Monitor (BAM1020, Met One Inc., US) using β-ray technology, and the meteorological parameters are obtained from a meteorological monitor (WXT520, VAISALA Inc., FL). The water-soluble inorganic ions and the gas species, including $NH_3$, $HNO_3$, and HCl, etc., are measured by a MARGA ion online analyzer (ADI2080, Metrohm Inc., CHN). Here the data from 2018 to 2023 are analyzed.

The thermodynamic model ISORROPIA v2.3 (Fountoukis and Nenes, 2007) is employed to predict the aerosol water content (AWC) and aerosol acidity, which is defined as the free molality of protons (Fountoukis and Nenes, 2007; Pye et al., 2020). Input parameters include $SO_4^{2-}$, total nitrate (gas $HNO_3$ + particle $NO_3^-$), total ammonia (gas $NH_3$ + particle $NH_4^+$), total chloride (gas HCl + particle $Cl^-$), NVCs and meteorological parameters like the temperature $T$ and relative humidity RH. The ISORROPIA model is run in the forward mode and metastable state 80 (Zheng et al., 2022b).

The ISORROPIA-predicted concentrations of $NH_3$, $NH_4^+$ and $NO_3^-$ agreed well with measurements ($R^2$ all above 0.95 and slopes all close to 1.0; Fig.S1). This demonstrate that thermodynamic analysis accurately reflects the aerosol state. However, the predicted $HNO_3$ concentration does not correlate well with the observed concentrations, as has been observed in many other studies (Ding et al., 2019; Zhou et al., 2022). This discrepancy may be 85 attributed to the high measurement uncertainty of gas-phase $HNO_3$ due to its low concentration (Rumsey et al., 2014).





**2.2 Time series analysis**

Time series analysis is a statistical method of analyzing a sequence of data points over an interval of time, which is particularly useful for understanding the structure and pattern of temporal data and is widely applied in

atmospheric studies(Shumway and Stoffer, 2017) (Hammer et al., 2020; Kang et al., 2020). Here, we performed time series analysis of pH and its potential influencing factors by decomposing them into 4 components: long-term trends, seasonal variations, diurnal cycles and random residues. Linear-fitting is adopted to predict the long-term trends (Kang et al., 2020; Mudelsee, 2019), and one-term Fourier curve fitting is adopted to fit the seasonal and diurnal cycles (Bloomfield, 2004; Singh et al., 2017). Here, we fixed the cycle period of Fourier curve as 1

year and 1 day in fitting the seasonal and diurnal variations, respectively. The random residues were obtained from the difference between the actual observed values and the sum of the predicted values of fitting functions in long-term trend, seasonal variations and diurnal cycles.

**2.3 Variation contribution quantification**

To quantify the contribution of a direct influencing factor to the variations of a certain term, the one-at-a-time

sensitivity analysis method is adopted (Yu et al., 2019). Briefly, assume variable $Y$ is a function of $n$ influencing factors of $x_1$ to $x_n$, i.e. $Y = f(x_1, …, x_n)$. The variations in $Y$ due to factor $x_i$, $\partial Y/\partial x_i$, is estimated as:

$$\partial Y/\partial x_i = f(\bar{x}_1, …, x_i, …, \bar{x}_n) - f(\bar{x}_1, …, \bar{x}_n) \tag{1}$$

where $x_i$ is the actual value of factor $x_i$, and $\bar{x}_i$ is the average of factor $x_i$. See more details in SI Text S1.

**3. Establishing the new hierarchical quantitative analysis framework**

**3.1 Interpretive structural model (ISM) based on multiphase buffer theory**

The recently proposed multiphase buffer theory reveals that most continental regions are within the ammonia-buffered regime, where the pH variations can be decomposed into(Zheng et al., 2020, 2022a):

$$pH = pK_a^* + c_{ni} + X_{gp} \tag{2a}$$
where

$$K_a^* = K_{a,NH_3} \frac{\rho_w}{H_{NH_3} R T \, AWC} \tag{2b}$$

$$c_{ni} = log \frac{\gamma_{H^+}}{\gamma_{NH_4^+}} \tag{2c}$$



$$X_{gp} = log \frac{[NH_3(g)]}{[NH_4^+(aq)]} \tag{2d}$$

Here, $K_a^*$ is the effective acid dissociation constant of $NH_3$ under ideal conditions in multiphase systems, $c_{ni}$ is the non-ideality correction factor, and $X_{gp}$ represents the gas-particle partitioning of $NH_3$. The $K_{a, NH3}$ is acid dissociation constant of $NH_3$ in bulk aqueous phase, $\rho_w$ is the water density, $H_{NH_3}$ is Henry's law constant of $NH_3$, $R$ is the gas constant, $T$ is temperature in K, and $\gamma_X$ is the activity coefficient of X.

Each term of the top-level pH decompositions (Eq. 2a) further depends on many other influencing factors, making the overall picture complicated. To illustrate the interconnections among these multiple driving factors, we applied the interpretive structural modeling (ISM) approach, which is widely used to identify and analyze the relationships between factors in complex systems (Sushil, 2012; Thakkar, 2021). With this method, a hierarchical relationship among influencing factors of aerosol pH can be established based on the multiphase buffer theory, as illustrated in Figure 1. Take the $pK_a^*$ for illustration. The $pK_a^*$ makes a direct impact on pH (top-level influencing factor), and its variation is determined by temperature and AWC. The AWC further depends mainly on $PM_{2.5}$ concentrations and RH, and minorly on the chemical profiles (middle level). Fundamentally, these influencing factors are caused by variations in synoptic conditions and emissions (bottom level).

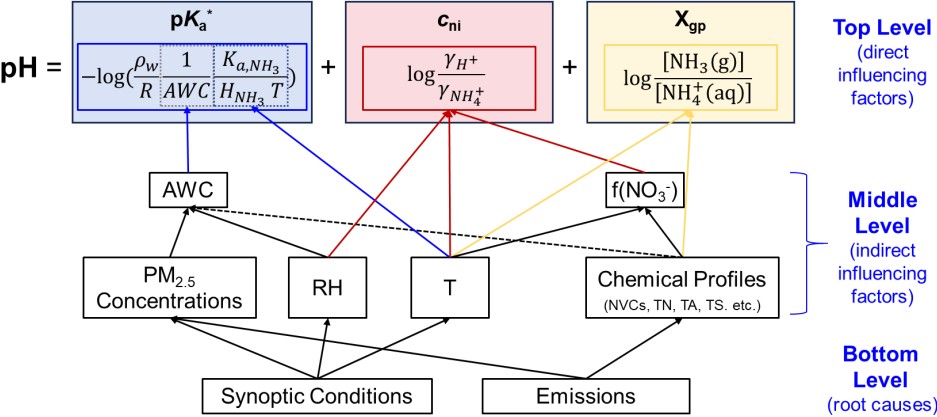

**Figure 1: Hierarchical relationship among influencing factors of aerosol pH based on the multiphase buffer theory as established with the interpretive structural modeling approach.**



### 3.2 ISM coupled with the time series analysis

With the above ISM model, a quantitative analysis of each factor following the influencing lines can be achieved. In addition, when coupled with the time series analysis, it can be applied to illustrate the driving factor of each time series component. Briefly, we can decompose each input parameter in ISORROPIA v2.3 into the 4 time series components (sect. 2.1 and 2.2), and then apply each component to explain upper-level factors of corresponding component. For example, the seasonal variations in Y due to seasonal variations of factor $x_i$, $\partial Y/\partial x_i |_{seas}$, is estimated as:

$$\partial Y/\partial x_i |_{seas} = f(\bar{x}_1, ..., x_{i, seas} + \bar{x}_i, ..., \bar{x}_n) - f(\bar{x}_1, ..., \bar{x}_n) \tag{3}$$

Where $x_{i, seas}$ is the decomposed seasonal variation of $x_i$. See more details in SI of Text S2.

### 4. Driving factor analysis of long-term data in Changzhou

Here we applied the new framework (sect. 3) to analyze the long-term data in Changzhou. From 2018 to 2023, around 90% periods are within the ammonia-buffered regime, while the rest are due to low RH ($< 0.3$) and aerosols not in a fully deliquescent state. Thus, the drivers of pH can be explained with the above framework.

The top-level ISM decomposition shows that the pH variations are mainly driven by the $pK_a^*$ (~52%) and $c_{ni}$ (~36%), while the $X_{gp}$ varies to a less extent (~12%, Figs. 2a and S2). In comparison, the time-series decomposition indicates that the pH variation is predominantly driven by seasonal variations and random residues, while the long-term trend and diurnal cycle play minor roles on the variations (Fig. 2b). Below we analyzed the driving factors of each time series component.

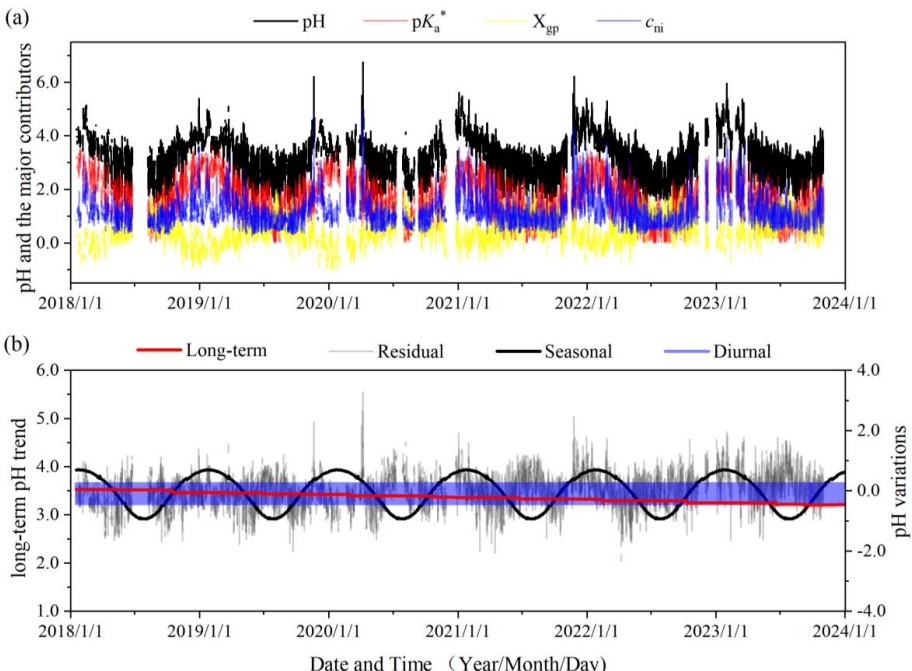

**Figure 2: Major components of pH variations.** (a) Decomposition into the $pK_a^*$, $X_{gp}$ and $c_{ni}$ base on the multiphase buffer theory. (b) Decomposition into long-term trends (left axis), seasonal variations, diurnal cycles and residuals (right axis) through time series analysis.

## 4.1 Long-term trends

The long-term pH trends in Changzhou show a slight decreasing trend of -0.05 year$^{-1}$ (Fig. 2b). The top-level ISM decompositions reveal that this is due to the competing trends of $pK_a^*$ and $c_{ni}$ with $X_{gp}$: while $pK_a^*$ and $c_{ni}$ decreased by -0.12 year$^{-1}$ and -0.14 year$^{-1}$, respectively, the $X_{gp}$ increased by 0.21 year$^{-1}$ (Fig. 3a).




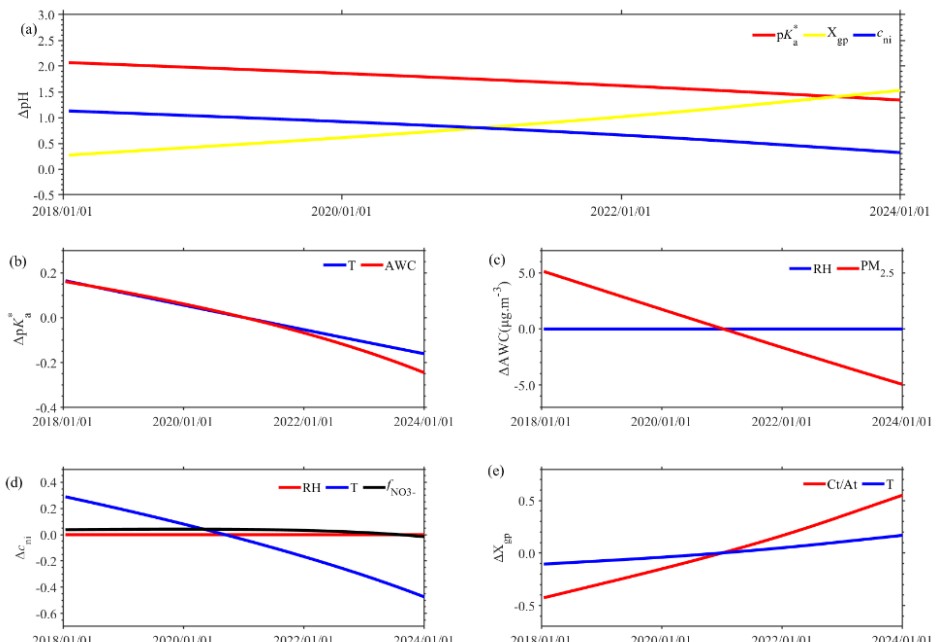

**Figure 3: Influencing factors of long-term pH variation at different levels.** (a) The 1st level decomposition into $pK_a^*$, $X_{gp}$ and $c_{ni}$. (b)-(e) Further investigation of the influencing factors of (b) $pK_a^*$ due to $T$ and AWC, (c) AWC due to RH and $PM_{2.5}$, (d) $c_{ni}$ due to RH, $T$, $f_{NO_3^-}$, and (e) $X_{gp}$ due to $C_t/A_t$ and $T$.

A further delve into the middle-level factors in the ISM reveals that the $pK_a^*$ decrease is due to the combined effect of decreasing AWC and increasing temperature (Fig. 3b). The temperature increased by 0.74 K·year$^{-1}$ (Fig. S3), corresponding to a $pK_a^*$ of -0.05 year$^{-1}$. In comparison, the AWC exhibited a decrease of -1.57 µg·m$^{-3}$·year$^{-1}$ (Fig. S3), corresponding to a $pK_a^*$ of -0.07 year$^{-1}$. The AWC decrease is primarily attributed to the $PM_{2.5}$ decrease (around -1.6 µg·m$^{-3}$·year$^{-1}$), while the long-term RH shows minimal variation (Fig. 3c). As to $c_{ni}$, its decreasing trend is mainly attributed to increased temperature, corresponding to $c_{ni}$ of -0.13 year$^{-1}$ (Fig. 3d). RH and $f_{NO_3^-}$ cause negligible effects on $c_{ni}$ because they were nearly constant (Fig. S3). In terms of $X_{gp}$, its increase is due to the increase in both relative abundance of alkaline to acidic substances ($C_t/A_t$) and temperature (Fig. 3e), contributing to the $X_{gp}$ increases of 0.16 year$^{-1}$ and 0.05 year$^{-1}$, respectively. Here the temperature influences $X_{gp}$ through the gas-particle partitioning volatility of semi-volatile species like ammonium nitrate. The increase in $C_t/A_t$ is further due to a much larger decrease in $A_t$ (sulfate, total nitrate, total chloride, etc.) than $C_t$ (total ammonia and NVCs, etc.) (Fig. S3).



Overall, we see that the long-term pH trend shows only a slight decrease despite considerable emission changes during this period, which is a combined effect of decreased $PM_{2.5}$ while increased temperature and $C_t/A_t$.

**4.2 Seasonal variations**

Influencing factors of seasonal variation pH are analyzed in similar ways with the long-term trends (**Error! Reference source not found.**). Overall, the pH is higher in winter and spring than summer and autumn, with the

amplitude of seasonal variations being 0.81, or the variation range being 1.62. The extent of variation is quantified by variation range hereinafter, which is the difference between the highest and lowest values for a given variable and for a given time series component. This cycle is consistent with $pK_a^*$ and $c_{ni}$ while in reverse phase with the $X_{gp}$.

The middle-level ISM decompositions demonstrate that the seasonal variation of $pK_a^*$ is mainly driven by the

temperature (Fig. 4b). The variation of temperature is 23.26 K (Fig. S4), which corresponds to a $pK_a^*$ variation of 1.20. In comparison, AWC varies to a less extent seasonally (Fig. S4), causing a relatively minor variation of 0.42 in $pK_a^*$. Seasonal variation of AWC is further primarily attributed to the variation in $PM_{2.5}$ levels (approximately 25.4 $\mu g \cdot m^{-3}$), as the seasonal RH varied little (Fig. 4c and Fig. S4). As to $X_{gp}$, its seasonal variation is influenced by both temperature and $C_t/A_t$, which correspond to the variations in $X_{gp}$ of -0.98 and -0.91, respectively (Fig. 4d).

Higher temperature and $C_t/A_t$ during summer facilitate more NHx to remain in the gas phase than winter (Fig. S4). Regarding $c_{ni}$, its seasonal variation is attributable to the combined effects of temperature and $f_{NO_3^-}$ (Fig.4e), leading to variations in $c_{ni}$ of 1.25 and 0.54, respectively. Again, influence of RH is negligible due to its little variations (Fig.4e). The seasonal variations of $f_{NO_3^-}$ is governed by significant variations of temperature (Fig. 4f). Temperature in winter is low enough and causes the vast majority of total nitrate to partition into the particle

phase, leading to minor variation of $f_{NO_3^-}$ with temperature variations (Fig.S5). Conversely, higher temperatures in summer result in more total nitrate existing as $HNO_3$ and $f_{NO_3^-}$ is sensitive to temperature variations.



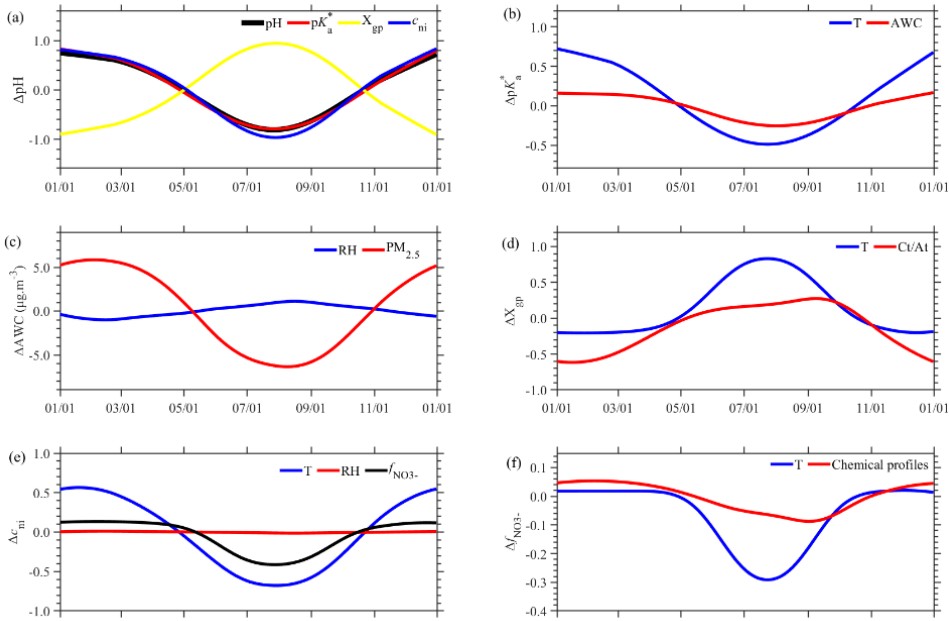

**Figure 4: Influencing factors of the seasonal variations of aerosol pH.** (a) The $1^{st}$ level decomposition into $pK_a^*$, $X_{gp}$ and $c_{ni}$. (b)-(f) Further investigation of the influencing factors of (b) $pK_a^*$ due to $T$ and AWC, (c) AWC due to RH and PM$_{2.5}$, (d)$X_{gp}$
due to $C_t/A_t$ and $T$, (e) $c_{ni}$ due to RH, $T$, $f_{NO_3^-}$, and (f) $f_{NO_3^-}$ due to $T$ and chemical profiles.

Overall, we see that the large seasonal variation of pH is mainly driven by the temperature as it plays a dominant

role in both $pK_a^*$, $c_{ni}$ and $X_{gp}$ (74%, 89% and 52%, respectively). In comparison, net influence of chemical profiles

is relatively smaller, contributing 48% and 10% to $X_{gp}$ and $c_{ni}$, respectively. That is, the seasonal variation of pH

is largely driven by the meteorology (esp. temperature) rather than emissions.

**4.3 Diurnal cycles**

Diurnal cycles of pH are higher at nighttime than daytime, with a variation range of 0.65. Similar to the seasonal

variations, the diurnal cycle is also consistent with $pK_a^*$ and $c_{ni}$ trend while in reverse trend with the $X_{gp}$ (Fig. S6a),

with their contribution to pH being 0.70, 0.58 and -0.63, respectively.

The major driving factors of diurnal cycles in $pK_a^*$ is different with that of seasonal variations. First, the diurnal

cycles of $pK_a^*$ is driven by both AWC (0.45) and temperature (0.25) (Fig. S6b), in contrast to the dominance of

temperature in seasonal variations (sect. 4.2). This is mainly due to the much smaller temperature variation range

diurnally (4.87 K; Fig. S7) than that seasonally (23.26 K). In addition, diurnal variation of AWC is primarily



influenced by RH (Fig. S6c) due to the larger RH diurnal variations (0.16 versus 0.03 in seasonal variations; Fig.

S7 and S4), in contrast with the PM$_{2.5}$ dominance in seasonal variations. In terms of X$_{gp}$, its dominant driving

factor of diurnal cycles is $C_t/A_t$, in contrast with the dominance of temperature in seasonal variations (Fig. S6d).

Diurnal cycles of $c_{ni}$ are due to the combined effects of temperature and $f_{NO_3^-}$ (Fig. S6e), where the $f_{NO_3^-}$ are

further mainly driven by chemical profiles (Fig. S6f).

### 4.4 Residual

The random residual of aerosol pH is another major contributor to pH temporal variations, which is comparable

with the seasonal variations. Distinct from all the three components above, the largest top-level contributor to

random residuals turns out to be $c_{ni}$ (82%), even exceeding that of p$K_a^*$ (42%), while X$_{gp}$ causes a net negative

effect (-24%; Fig. S8). Moreover, random fluctuations of $c_{ni}$ are almost entirely due to variations in $f_{NO_3^-}$ (~95%;

Fig. 5a), which are primarily driven by chemical profiles (~93%; Fig. 5b). The p$K_a^*$ random fluctuations are

mainly caused by AWC (~79%), which is further attributed mainly (~60%) to PM$_{2.5}$ variations (Fig. 5d). Random

variations of X$_{gp}$ are dominated by the chemical profiles, similar to diurnal cycles and long-term trend (Fig.S9).

Overall, PM$_{2.5}$ and chemical profiles are the major influencing factors for random residual of pH, underscoring

the prominence of emissions over meteorology.



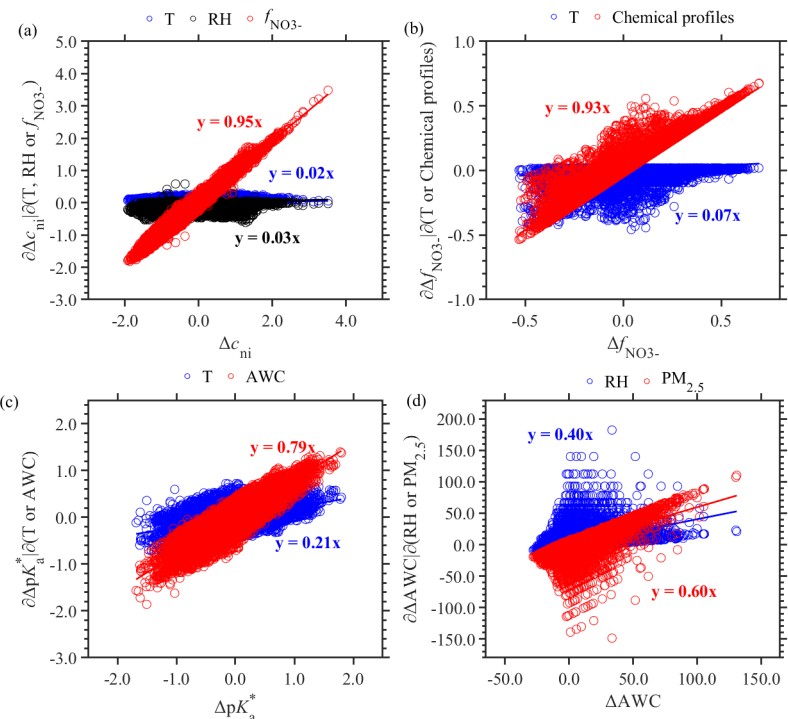

**Figure 5: Influencing factors of the random residual of aerosol pH.** (a) $c_{ni}$ due to $T$, RH, and $f_{NO_3^-}$, (b) $f_{NO_3^-}$ due to $T$ and Chemical profiles, (c) $pK_a^*$ due to $T$ and AWC, (d) AWC due to RH and $PM_{2.5}$.

## 5. Overall contributions and implications

Figure 6 shows the distinct major influencing factors of aerosol pH in the 4 time series components, where the factors contributing less than |10%| are not shown. Overall, $pK_a^*$ is the dominant influencing factor of pH variations, being the major contributor in all components, and playing a pivotal role in seasonal and diurnal cycles (Fig. 6). The $c_{ni}$ is another dominant factor for pH variations, especially in random fluctuations. The $X_{gp}$ shows a reverse trend with pH in all components. As seasonal and random variations largely regulate the pH temporal variations, the $pK_a^*$ and $c_{ni}$ contribute more to pH than $X_{gp}$.



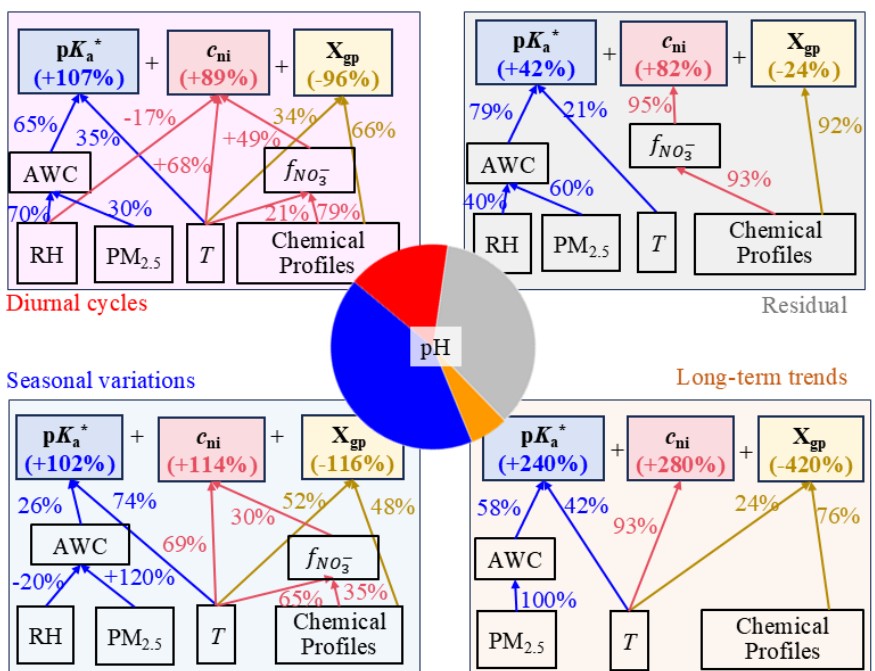

**Figure 6: Hierarchical relationship among major influencing factors of aerosol pH during 4 time series** (Factors contributing less than |10%| are not shown).

Deeper-level driver analysis show that meteorology plays a more important role than chemical profiles in explaining the pH temporal variations (57% versus 22%; Fig. S10). Temperature is the major contributor, which explains seasonal variations in $pK_a^*$, $c_{ni}$, and $X_{gp}$ of 74%, 89% and 52%, respectively, and it is also important in diurnal cycles and long-term trends. RH plays an important role in diurnal cycles, accounting for 70% of the AWC diurnal variations and thus indirectly exerting an important effect on $pK_a^*$ (~46%). In comparison, chemical profiles are essential for explaining long-term trends in $X_{gp}$, and they also provide a pivotal role in random fluctuations and diurnal cycles for $c_{ni}$. The PM$_{2.5}$ concentration is an overall effect of meteorology and emissions. PM$_{2.5}$ is the dominant contributor to AWC in all components except diurnal cycles, exerting an indirect influence on $pK_a^*$. Overall, temperature is critical in explaining pH variations (48%; Fig. S10), followed by chemical profiles, PM$_{2.5}$ concentrations and RH (22%, 21% and 9%, respectively).

The quantitative framework we proposed here can provide a clear understanding of the drivers of aerosol acidity temporal variations, with information on both quantitative contributions and the underlying mechanisms. Our



findings suggest that relative importance of synoptic conditions versus emissions in aerosol acidity variations

differed much with the time scale of concern and are through different major mechanisms. In Changzhou, synoptic

conditions are more important for seasonal variations and diurnal cycles of pH, while emissions cause greater

effect on pH random fluctuations. For the long-term trends, both emissions and synoptic conditions are important.

In other places, this framework still applies, while the conclusions may vary. These quantitative understandings

on the driving factors of aerosol acidity are important in acidity-relevant process studies and policy-making, such

as nitrate control(Guo et al., 2017), sulfate formation(Cheng et al., 2016; Zheng et al., 2024a), nitrogen depositions

(Nenes et al., 2021), etc.

**Data availability.** The dataset of aerosol chemical components and precursor gases is available on the website
http://192.168.100.25/czems/MainFrame.aspx supported by Changzhou Ecological Environment Bureau.

**Author contributions.** G.Z. and Q. Zhang designed and led the study. X. D., G.Z. and C. Chen performed the
study. C. Chen conducted the campaign and collected the data. X. D. and G.Z. wrote the manuscript with the input
of all authors. All authors have given approval to the final version of the manuscript.

**Competing interests.** At least one of the (co-)authors is a member of the editorial board of Atmospheric
Chemistry and Physics.

**Acknowledgments.** The research was supported by the National Natural Science Foundation of China (22188102).
C. Chen thanks the Shuimu Tsinghua Scholar Program (2023SM027).

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
