# Peer review of "Driving factors of aerosol acidity: a new hierarchical quantitative analysis framework and its application in Changzhou, China"

_EGUsphere, 2024_

## Author Comment (AC1)

**Title:** "Driving factors of aerosol acidity: a new hierarchical quantitative analysis framework and its application in Changzhou, China"

**Manuscript No.:** egusphere-2024-3584

**Authors:** Duan et al.

*General Comments*:

*Aerosol acidity is one of the core parameters in atmospheric chemistry. In recent years, much interesting work has been done on the trends and driving factors of aerosol acidity changes. This article is pioneering in decomposing the trends of aerosol acidity changes into long-term, seasonal, diurnal, and random components, and decoupling the driving factors into meteorological and emission drivers. This research framework greatly simplifies the interpretation of the results from complex multiphase buffering theory. I believe that with minor revisions, this article can be published in Atmospheric Chemistry and Physics.*

*Here are my specific recommendations:*

**Responses:**

We thank the referee for the valuable and constructive comments/suggestions on our manuscript. We have revised the manuscript accordingly, and please find our point-to-point responses below.

*Specific comments:*

*1.Lines 35-36: The expression here is not very precise. Andrew Ault et al have focused on directly measuring aerosol pH. Although their methods have limitations and have not yet been widely applied in practical measurements, direct measurement methods do exist.*

**Responses:**

We thank the reviewer for the comment. We've modified the expressions as:

"Current direct measurement methods of aerosol pH (Ault, 2020) are not yet applied in ambient observations due to limitations such as slow measurement speeds (Lei et al., 2020) or the targeting of single particles (Craig et al., 2017). Therefore, thermodynamic models are widely adopted to estimate aerosol acidity and investigate its influencing factors (Clegg et al., 2001; Fountoukis and Nenes, 2007; Tao and Murphy, 2021; Zaveri et al., 2008; Zuend et al., 2008)."

*2.Lines 57-64: I hope to use highly concise language to summarize the differences between this study and previous research on chemical profiles and meteorological parameters driven pH changes, as well as highlight the most innovative and distinctive features of this article.*

**Responses:**

We thank the reviewer for the comment. We've added this point as (see the last paragraph in Sect. 1):

"Combining this model with time series analysis, we proposed a novel hierarchical quantitative analysis framework, which can not only quantify the contribution of different influencing factors, but also reveal the underlying mechanisms and dominant pathways of the influences. Compared with previous studies, this framework can provide a more systematic, in-depth and quantitative understanding on how the meteorology or chemical profiles would affect aerosol pH over different time scales of interest. Applying this framework to the long-term observations in Changzhou, China, distinct driving factors and underlying mechanisms were quantified for different time series components, and future implications were also discussed."

*3.Line 87: Can mathematical formulas be provided here? For example, linear fitting and Fourier curve fitting, as well as how to use mathematical methods to decouple the trends of 4 components.*

**Responses:**

We thank the reviewer for the comment. We've added this point as (see Sect. 2.2 and Text S1 in SI):

"Linear-fitting ($y = a*t + b$, where $t$ is defined as time hereinafter) is adopted to predict the long-term trends (Kang et al., 2020; Mudelsee, 2019), and one-term Fourier curve fitting ($y = a_0 + a_1*\cos(\omega*t) + b_1*\sin(\omega*t)$) is adopted to fit the seasonal and diurnal cycles (Bloomfield, 2004; Singh et al., 2017)."

Sect 2.2, Line 105 in the revised manuscript: "See more details in SI Text S1-S2"

**"S1. Detailed description of decomposing parameters into 4 time series components**

The decomposition process consists of the following main steps: linear-fitting of the long-term trends, one-term Fourier curve fitting of seasonal and diurnal variations, and extraction of random residues. For parameter *p*, this process is expressed as Eq. S1a-d:

$$p_{yr} = a_1*t + b_1 \tag{S1a}$$

$$p_{seas} = a_2 + b_2*\sin(\omega_1*t) + c_1 \times \cos(\omega_1*t) \tag{S1b}$$

$$p_{day} = a_3 + b_3*\sin(\omega_2*t) + c_2 \times \cos(\omega_2*t) \tag{S1c}$$

$$p_{res} = p - p_{yr} - p_{seas} - p_{day} \tag{S1d}$$

where $p$, $p_{yr}$, $p_{seas}$, $p_{day}$ and $p_{res}$ are the values of actual observed, long-term trend, seasonal variation, diurnal cycle and residues, respectively. The $t$ is the time, and $a_i$, $b_i$

and $c_i$ are the coefficients of fitted curves during the corresponding time series, respectively. $\omega_1$ and $\omega_2$ are $2\pi/365$ days$^{-1}$ and $2\pi/24$ hours$^{-1}$, respectively, to fixed the cycle period of Fourier curve as 1 year and 1 day in fitting the seasonal and diurnal variations."

*4.Line 142: It is recommended to use percentiles for RH.*

**Responses:**

We thank the reviewer for the comment. We've used percentiles for RH throughout manuscript (see the paragraph before Sect. 4.1 and second paragraph in Sect. 4.3), and we also modified the axis labels in figures about RH (Fig. S4d and S7d).

*5.Lines 144-145: The decomposition of pH into the three factors can be understood mathematically. However, is it appropriate to plot these three factors as time series? From the perspective of aerosol physicochemical properties, especially the meaning of $H^+$, plotting them as time series may not be easily interpretable.*

**Responses:**

We thank the reviewer for the comment. We've adopted a bottom-up method to quantify the time series components of pH. Each input parameter $p$ in ISORROPIA v2.3 is subdivided into 4 time series components, which are further used in ISORROPIA v2.3 to obtain the pH at corresponding time series. We've performed time-series decomposition of all influencing factors. This process clarifies how the fluctuations of different factors affect pH levels. The underlying principle of such decompositions in time series analysis is that some influencing factors present periodic variation, with the most common period being seasons or days (Anderson, 2011; Wei, 2013). For example, one important influencing factor of aerosol acidity, the temperature, exhibits significant

seasonal and diurnal variations; in comparison, emission profile of some species may show relatively small diurnal variations but stronger seasonally variations. Thus, we consider it reasonable to plot these factors as a time series. We've clarified these points in the revised manuscript and SI text S2 as:

Sect 2.2, Line 105 in the revised manuscript: "See more details in SI Text S1-S2".

"S2. Detailed descriptions of quantitative analysis of each factor based on ISM and time series analysis

Here we adopted a bottom-up method to quantify the time series components of upper-level factors in the ISM model and its driving factors. That is, based on the decomposition of time series analysis, each input parameter $p$ in ISORROPIA v2.3 is subdivided into 4 time series components. The underlying principle of such decompositions is that most influencing factors of aerosol acidity, such as temperature and emissions, are influenced by long-term variations, periodical variations (i.e., seasonally and diurnally) and random fluctuations (Anderson, 2011; Wei, 2013). For example, temperature can be decomposed into long-term trend ($T_{yr}$), seasonal variations ($T_{seas}$), diurnal cycles ($T_{day}$) and residuals ($T_{res}$), respectively."

*6. Lines 149-152: The colors are unclear. It is recommended to increase the thickness of the lines in the legend.*

**Responses:**

We thank the reviewer for the comment. We've increased the thickness of the lines in the legend of Fig. 2.

*7. Lines 178-179: There seems to be an error here.*

**Responses:**

We're sorry for this mistake. There is a citation of Fig. 4, and we've corrected it.

125

*8.Lines 239-241: What are the percentage changes in pKa\*, $c_{ni}$ and $X_{gp}$ relative to? The difference in pH or the original pH? This could be expressed more clearly in the figure caption.*

**Responses:**

130 We thank the reviewer for the comment. The percentage variations in *pKa\*, $c_{ni}$ and $X_{gp}$* is relative to the pH variations. We've clarified this point in the figure caption of Fig. 6 as:

"**Figure 6: Hierarchical relationship among major influencing factors of aerosol pH variations for the 4 time series components, respectively.** Here, the percentage

135 variations are derived by the variations due to factor X to overall variations. For example, the contribution of $pK_a^*$ variations to seasonal variations of pH is derived by $\Delta pK_a^*,_{seas} / \Delta pH,_{seas} * 100\%$, where the overall pH variations $\Delta pH,_{seas} = \Delta pK_a^*,_{seas} + \Delta c_{ni, seas} + \Delta X_{gp, seas}$. Factors contributing less than |10%| are not shown."

140 **References**

Anderson, T. W.: The statistical analysis of time series, John Wiley & Sons, 2011.

Ault, A. P.: Aerosol Acidity: Novel Measurements and Implications for Atmospheric Chemistry, Acc. Chem. Res., 53, 1703–1714, https://doi.org/10.1021/acs.accounts.0c00303, 2020.

145 Bloomfield, P.: Fourier Analysis of Time Series: An Introduction, John Wiley & Sons, 285 pp., 2004.

Clegg, S. L., Seinfeld, J. H., and Brimblecombe, P.: Thermodynamic modelling of aqueous aerosols containing electrolytes and dissolved organic compounds, Journal of Aerosol Science, 32, 713–738, https://doi.org/10.1016/S0021-
150 8502(00)00105-1, 2001.

Craig, R. L., Nandy, L., Axson, J. L., Dutcher, C. S., and Ault, A. P.: Spectroscopic

Determination of Aerosol pH from Acid–Base Equilibria in Inorganic, Organic, and Mixed Systems, J. Phys. Chem. A, 121, 5690–5699, https://doi.org/10.1021/acs.jpca.7b05261, 2017.

155 Fountoukis, C. and Nenes, A.: ISORROPIA II: a computationally efficient thermodynamic equilibrium model for K +–Ca2+–Mg2+–NH+ 4 –Na+–SO2− 4 –NO− 3 –Cl−–H2O aerosols, Atmos. Chem. Phys., 4639–4659, 2007.

Kang, Y.-H., You, S., Bae, M., Kim, E., Son, K., Bae, C., Kim, Y., Kim, B.-U., Kim, H. C., and Kim, S.: The impacts of COVID-19, meteorology, and emission control
160 policies on PM2.5 drops in Northeast Asia, Sci Rep, 10, 22112, https://doi.org/10.1038/s41598-020-79088-2, 2020.

Lei, Z., Bliesner, S. E., Mattson, C. N., Cooke, M. E., Olson, N. E., Chibwe, K., Albert, J. N. L., and Ault, A. P.: Aerosol Acidity Sensing via Polymer Degradation, Anal. Chem., 92, 6502–6511, https://doi.org/10.1021/acs.analchem.9b05766, 2020.

165 Mudelsee, M.: Trend analysis of climate time series: A review of methods, Earth-Science Reviews, 190, 310–322, https://doi.org/10.1016/j.earscirev.2018.12.005, 2019.

Singh, P., Joshi, S. D., Patney, R. K., and Saha, K.: The Fourier decomposition method for nonlinear and non-stationary time series analysis, Proc. R. Soc. A., 473,
170 20160871, https://doi.org/10.1098/rspa.2016.0871, 2017.

Tao, Y. and Murphy, J. G.: Simple Framework to Quantify the Contributions from Different Factors Influencing Aerosol pH Based on NHx Phase-Partitioning Equilibrium, Environ. Sci. Technol., 55, 10310–10319, https://doi.org/10.1021/acs.est.1c03103, 2021.

175 Wei, W. W. S.: The Oxford Handbook of Quantitative Methods, Vol. 2: Statistical Analysis, Oxford University Press, https://doi.org/10.1093/oxfordhb/9780199934898.001.0001, 2013.

Zaveri, R. A., Easter, R. C., Fast, J. D., and Peters, L. K.: Model for Simulating Aerosol Interactions and Chemistry (MOSAIC), J. Geophys. Res., 113, 2007JD008782,
180 https://doi.org/10.1029/2007JD008782, 2008.

Zuend, A., Marcolli, C., Luo, B. P., and Peter, T.: A thermodynamic model of mixed organic-inorganic aerosols to predict activity coefficients, Atmos. Chem. Phys., 2008.

185

---

## Author Response (AR1)

**Title:** "Driving factors of aerosol acidity: a new hierarchical quantitative analysis framework and its application in Changzhou, China"

**Manuscript NO.:** egusphere-2024-3584

**Authors:** Duan et al.

We thank the referees for their valuable and constructive comments/suggestions on our manuscript. We have revised the manuscript accordingly and please find our point-to-point responses below.

*Comments by Anonymous Referee #2*

*General Comments*:

*Understanding the driving factors of aerosol pH is important, as aerosol pH drives many multiphase processes in the atmosphere. Duan et al. developed a new framework to decouple the main factors that influence aerosol pH. The results have important*

*implications for atmospheric chemistry and air quality studies. The manuscript can be recommended for publication after the following concerns are addressed.*

**Responses:**

We thank the reviewer for the positive comments. Please find our point-to-point responses below.

*Specific comments:*

*1. Lines 84-86: As shown in Fig. S1, the highest gas-phase $HNO_3$ concentration was around 6 µg/m³, which is not a low value. Are there any other reasons leading to the*

*measurement uncertainty of HNO₃? A related question is how the uncertainty of HNO₃*

*would bias the main conclusion.*

**Responses:**

We thank the reviewer for the comment. In addition to low concentrations, the viscosity properties of $HNO_3$ may cause its adsorption on the MARGA units' inlet and tubing, which can also affect the measurement uncertainty (Rumsey et al., 2014). The discrepancy between observed and predicted $HNO_3$ concentrations may also be attributed to the calculation errors in the ISORROPIA thermodynamic equilibrium model, particularly in the prediction of activity coefficients (Zheng et al., 2022). The influence of this discrepancy in Changzhou is expected to be small, as indicated by the good agreement between observed and predicted $NH_3(g)$ and $NH_4^+$ (Fig. S1), and the fraction of $NO_3^-$ in particle-phase anions $f_{NO3^-}$ (y = 0.94x, $R^2$ = 0.95). In other scenarios, this discrepancy may be more important, which needs further evaluations in future studies. Nevertheless, it would not influence the applicability of the framework we proposed here. We've added these points as (see last paragraph in Sect. 2.1):

"This discrepancy may be attributed to two factors: (1) the high measurement uncertainty of gas-phase $HNO_3$ due to its low concentration and viscous properties, which may cause its adsorption on the MARGA units' inlet and tubing (Rumsey et al., 2014); and (2) the calculation accuracy of ISORROPIA thermodynamic equilibrium model, particularly in predicting activity coefficients (Zheng et al., 2022). The influence of this discrepancy on the results in this study is tested to be small, but its influence under other scenarios needs to be examined in future studies."

*2.Lines 90-92: Does this mean that the sum of long-term trend, season variation, diurnal cycle, and random residues is aerosol pH? Can the authors elaborate on the rationale?*

**Responses:**

We thank the reviewer for the comment. Yes, the sum of 4 time series values of pH is the actual aerosol pH. This is technically ensured as that the random residue is obtained by:

Random residue = actual aerosol pH – long-term trend – seasonal variation – diurnal cycles

The underlying principle of such decompositions in time series analysis is that some influencing factors present a regular periodic variation across the time series, with the most common period being seasons or days (Anderson, 2011; Wei, 2013). For example, one important influencing factor of aerosol acidity, the temperature, would show obvious seasonal and diurnal variations, despite the random residues due to specific weather conditions. In comparison, the emission profile of some species may show relatively small diurnal variations but stronger seasonally variations. Here, by decomposing the influencing factors into these four components, we aim to identify the dominant periodical fluctuations in addition to the contribution of each influencing factors.

We've clarified these points in the revised manuscript and SI text S2 as:

Sect 2.2, Line 105 in the revised manuscript: "See more details in SI Text S1-S2"

**"S2. Detailed descriptions of quantitative analysis of each factor based on ISM and time series analysis**

Here we adopted a bottom-up method to quantify the time series components of upper-level factors in the ISM model and its driving factors. That is, based on the decomposition of time series analysis, each input parameter p in ISORROPIA v2.3 is subdivided into 4 time series components. The underlying principle of such decompositions is that most influencing factors of aerosol acidity, such as temperature and emissions, are influenced by long-term variations, periodical variations (i.e., seasonally and diurnally) and random fluctuations (Anderson, 2011; Wei, 2013). For example, temperature can be decomposed into long-term trend ($T_{yr}$), seasonal variations ($T_{seas}$), diurnal cycles ($T_{day}$) and residuals ($T_{res}$), respectively."

*3.Lines 120-125: Please elaborate on the connection between middle-level and top-level factors. For example, how $f(NO_3^-)$ would affect $C_{ni}$ is unclear.*

**Responses:**

We thank the reviewer for the comment. We've clarified the connection between each level factors in ISM as (see last paragraph in Sect.3.1 in main text and Text S4 in SI):

"Take the p$K_a^*$ for illustration here; other direct drivers of $c_{ni}$ and $X_{gp}$ are elaborated in SI Text S4 (Zheng et al., 2022, 2024)."

**"S4. Detailed descriptions of hierarchical relationship between influencing factors for $c_{ni}$ and $X_{gp}$ in ISM**

The $c_{ni}$ causes direct effects on aerosol pH (Top-level). As shown in Eq. 2c, $c_{ni}$ is the
ratio of activity coefficients, which firstly depends on RH and Temperature (middle-level). The $NO_3^-$ and $SO_4^{2-}$ are the main anions paired with $H^+$ or $NH_4^+$ in ammonia-buffered ambient aerosols, and it has been shown that $c_{ni}$ differs between sulfate- and nitrate-dominated aerosols (Zheng et al., 2022). The $c_{ni}$ at a given RH and temperature therefore depends on the anion profiles, which can be expressed as the fraction of $NO_3^-$
in anions (aq), i.e., $f_{NO3-}$ (middle-level) (Zheng et al., 2022). The $f_{NO3-}$ is further sensitive to temperature, which influences the gas-particle partitioning of volatility of semi-volatile species like ammonium nitrate, and chemical profiles (middle-level). Meteorological parameters and chemical profiles are fundamentally related to synoptic conditions and emissions (bottom-level).

$X_{gp}$ also directly affects aerosol pH (top-level), and its variation is influenced by temperature and chemical profiles (middle-level) (Zheng et al., 2024). Similar with p$K_a^*$

and $c_{ni}$, the root of these variations is synoptic conditions and emissions (bottom-level)."

*4.Line 259: Can the authors provide more comparisons with other studies in the YRD region?*

**Responses:**

We thank the reviewer for the comment. There were some studies in the YRD region proved that acidic anions and NVCs are the primary drivers for pH long-term trends, which is one of the factors in our findings (Lv et al., 2024; Zhou et al., 2022). While Lv et al. (2024) emphasized the role of chemical profiles, the importance of temperature in the pH seasonal variations is well recognized in these previous studies. For diurnal cycles of pH, both studies underscored the crucial influence of meteorological conditions. Overall, most of our findings are consistent with previous studies in the YRD region. We've added more comparisons with other studies in the YRD region as:

In Changzhou, synoptic conditions are more important for seasonal variations and diurnal cycles of pH, while emissions cause greater effect on pH random fluctuations. For the long-term trends, both emissions and synoptic conditions are important. These findings generally agree with previous studies in the YRD region. For example, two previous studies on aerosol acidity in Shanghai (Lv et al., 2024; Zhou et al., 2022) also found that synoptic conditions played important role in seasonal and diurnal variations of aerosol pH, while for long-term pH trends, the primary drivers were attributed to emissions in acidic anions and NVCs. Regardless, our analysis here provided more systematic insight and theoretically explanations into the underlying mechanisms of each influencing factor compared with previous studies. In other places, this framework still applies, while the conclusions may vary."

*Technical comments:*

*1.Lines 41-42: Please correct the reference format.*

**Responses:**

We thank the reviewer for the comment. We've corrected this point and checked throughout.

*2.Line 43: "which" should be "while".*

**Responses:**

We thank the reviewer for the comment. We've corrected accordingly and checked throughout.

*3.Line 179: Please correct the reference format.*

**Responses:**

We're sorry for this mistake. There is a citation of Fig. 4 rather than a reference, and we've corrected it.

*4.The quality of the figures should be improved.*

**Responses:**

We thank the reviewer for the comment. We've improved the quality of the figures throughout.

*Comments by Anonymous Referee #3*

*General Comments*:

*Aerosol acidity is one of the core parameters in atmospheric chemistry. In recent years, much interesting work has been done on the trends and driving factors of aerosol acidity changes. This article is pioneering in decomposing the trends of aerosol acidity changes into long-term, seasonal, diurnal, and random components, and decoupling the driving factors into meteorological and emission drivers. This research framework greatly*

*simplifies the interpretation of the results from complex multiphase buffering theory. I believe that with minor revisions, this article can be published in Atmospheric Chemistry and Physics.*

*Here are my specific recommendations:*

**Responses:**

We thank the reviewer for the positive comments. Please find our point-to-point responses below.

*Specific comments:*

1.*Lines 35-36: The expression here is not very precise. Andrew Ault et al have focused*

*on directly measuring aerosol pH. Although their methods have limitations and have not yet been widely applied in practical measurements, direct measurement methods do exist.*

**Responses:**

We thank the reviewer for the comment. We've modified the expressions as:

"Current direct measurement methods of aerosol pH (Ault, 2020) are not yet applied in ambient observations due to limitations such as slow measurement speeds (Lei et al., 2020) or the targeting of single particles (Craig et al., 2017). Therefore, As direct

 thermodynamic models are widely adopted to estimate aerosol acidity and investigate its influencing factors (Clegg et al., 2001; Fountoukis and Nenes, 2007; Tao and Murphy, 2021; Zaveri et al., 2008; Zuend et al., 2008)."

*2.Lines 57-64: I hope to use highly concise language to summarize the differences between this study and previous research on chemical profiles and meteorological parameters driven pH changes, as well as highlight the most innovative and distinctive features of this article.*

**Responses:**

We thank the reviewer for the comment. We've added this point as (see the last paragraph in Sect. 1):

"Combining this model with time series analysis, we proposed a  novel hierarchical quantitative analysis framework , which can not only quantify the contribution of different influencing factors, but also reveal the underlying mechanisms and dominant pathways of the influences. Compared with previous studies, this framework can provide a more systematic, in-depth and quantitative understanding on how the meteorology or chemical profiles would affect aerosol pH over different time scales of interest.  Applying this framework to the long-term observations in Changzhou, China distinct driving factors and underlying mechanisms were  quantified for  different time series components,  and future implications were also discussed."

*3.Line 87: Can mathematical formulas be provided here? For example, linear fitting and Fourier curve fitting, as well as how to use mathematical methods to decouple the*

*trends of 4 components.*

**Responses:**

We thank the reviewer for the comment. We've added this point as (see Sect. 2.2 and Text S1 in SI):

"Linear-fitting (y=a$*t$ + b, where $t$ is defined as time hereinafter) is adopted to predict the long-term trends (Kang et al., 2020; Mudelsee, 2019), and one-term Fourier curve fitting (y=a$_0$+a$_1*$cos($\omega*t$) +b$_1*$sin($\omega*t$)) is adopted to fit the seasonal and diurnal cycles (Bloomfield, 2004; Singh et al., 2017)."

Sect 2.2, Line 105 in the revised manuscript: "See more details in SI Text S1-S2"

"**S1. Detailed description of decomposing parameters into 4 time series components**

The decomposition process consists of the following main steps: linear-fitting of the long-term trends, one-term Fourier curve fitting of seasonal and diurnal variations, and extraction of random residues. For parameter $p$, this process is expressed as Eq. S1a-d:

$$p_{yr} = a_1*t + b_1 \tag{S1a}$$

$$p_{seas} = a_2 + b_2*\sin(\omega_1*t) + c_1 \times \cos(\omega_1*t) \tag{S1b}$$

$$p_{day} = a_3 + b_3*\sin(\omega_2*t) + c_2 \times \cos(\omega_2*t) \tag{S1c}$$

$$p_{res} = p - p_{yr} - p_{seas} - p_{day} \tag{S1d}$$

where $p$, $p_{yr}$, $p_{seas}$, $p_{day}$ and $p_{res}$ are the values of actual observed, long-term trend, seasonal variation, diurnal cycle and residues, respectively. The $t$ is the time, and $a_i$, $b_i$ and $c_i$ are the coefficients of fitted curves during the corresponding time series, respectively.

respectively. $\omega_1$ and $\omega_2$ are $2\pi/365$ days$^{-1}$ and $2\pi/24$ hours$^{-1}$, respectively, to fixed the cycle period of Fourier curve as 1 year and 1 day in fitting the seasonal and diurnal variations."

*4.Line 142: It is recommended to use percentiles for RH.*

**Responses:**

We thank the reviewer for the comment. We've used percentiles for RH throughout manuscript (see the paragraph before Sect. 4.1 and second paragraph in Sect. 4.3), and we also modified the axis labels in figures about RH (Fig. S4d and S7d).

*5.Lines 144-145: The decomposition of pH into the three factors can be understood mathematically. However, is it appropriate to plot these three factors as time series? From the perspective of aerosol physicochemical properties, especially the meaning of $H^+$, plotting them as time series may not be easily interpretable.*

**Responses:**

We thank the reviewer for the comment. We've adopted a bottom-up method to quantify the time series components of pH. Each input parameter $p$ in ISORROPIA v2.3 is subdivided into 4 time series components, which are further used in ISORROPIA v2.3 to obtain the pH at corresponding time series. We've performed time-series decomposition of all influencing factors. This process clarifies how the fluctuations of
different factors affect pH levels. The underlying principle of such decompositions in time series analysis is that some influencing factors present periodic variation, with the most common period being seasons or days (Anderson, 2011; Wei, 2013). For example, one important influencing factor of aerosol acidity, the temperature, exhibits significant seasonal and diurnal variations; in comparison, emission profile of some species may
show relatively small diurnal variations but stronger seasonally variations. Thus, we consider it reasonable to plot these factors as a time series. We've clarified these points in the revised manuscript and SI text S2 as:

Sect 2.2, Line 105 in the revised manuscript: "See more details in SI Text S1-S2"

"S2. Detailed descriptions of quantitative analysis of each factor based on ISM and time series analysis

Here we adopted a bottom-up method to quantify the time series components of upper-level factors in the ISM model and its driving factors. That is, based on the decomposition of time series analysis, each input parameter $p$ in ISORROPIA v2.3 is subdivided into 4 time series components. The underlying principle of such decompositions is that most influencing factors of aerosol acidity, such as temperature and emissions, are influenced by long-term variations, periodical variations (i.e., seasonally and diurnally) and random fluctuations (Anderson, 2011; Wei, 2013). For example, temperature can be decomposed into long-term trend ($T_{yr}$), seasonal variations ($T_{seas}$), diurnal cycles ($T_{day}$) and residuals ($T_{res}$), respectively."

*6.Lines 149-152: The colors are unclear. It is recommended to increase the thickness of the lines in the legend.*

**Responses:**

We thank the reviewer for the comment. We've increased the thickness of the lines in the legend of Fig. 2.

*7.Lines 178-179: There seems to be an error here.*

**Responses:**

We're sorry for this mistake. There is a citation of Fig. 4, and we've corrected it.

*8.Lines 239-241: What are the percentage changes in pKa\*, cni and Xgp relative to? The difference in pH or the original pH? This could be expressed more clearly in the figure caption.*

**Responses:**

We thank the reviewer for the comment. The percentage variations in $pKa^*$, $c_{ni}$ and $X_{gp}$ is relative to the pH variations. We've clarified this point in the figure caption of Fig. 6 as:

"**Figure 6: Hierarchical relationship among major influencing factors of aerosol pH variations for the 4 time series components, respectively.** Here, the percentage variations are derived by the variations due to factor X to overall variations. For example, the contribution of $pK_a^*$ variations to seasonal variations of pH is derived by $\Delta pK_a^*_{,\ seas} / \Delta pH_{,\ seas} * 100\%$, where the overall pH variations $\Delta pH_{,\ seas} = \Delta pK_a^*_{,\ seas} + \Delta c_{ni,\ seas} + \Delta X_{gp,\ seas}$. Factors contributing less than $|10\%|$ are not shown."

**References**

Anderson, T. W.: The statistical analysis of time series, John Wiley & Sons, 2011.

[revised manuscript text omitted]